# Why rankings of biomedical image analysis competitions should be interpreted with care

Lena Maier-Hein ⓘ et al.[#]

International challenges have become the standard for validation of biomedical image analysis methods. Given their scientific impact, it is surprising that a critical analysis of common practices related to the organization of challenges has not yet been performed. In this paper, we present a comprehensive analysis of biomedical image analysis challenges conducted up to now. We demonstrate the importance of challenges and show that the lack of quality control has critical consequences. First, reproducibility and interpretation of the results is often hampered as only a fraction of relevant information is typically provided. Second, the rank of an algorithm is generally not robust to a number of variables such as the test data used for validation, the ranking scheme applied and the observers that make the reference annotations. To overcome these problems, we recommend best practice guidelines and define open research questions to be addressed in the future.

Biomedical image analysis has become a major research field in biomedical research, with thousands of papers published on various image analysis topics including segmentation, registration, visualization, quantification, object tracking, and detection[1,2]. For a long time, validation and evaluation of new methods were based on the authors' personal data sets, rendering fair and direct comparison of the solutions impossible[3]. The first known efforts to address this problem date back to the late 90 s[4], when Jay West, J Michael Fitzpatrick and colleagues performed an international comparative evaluation on intermodality brain image registration techniques. To ensure a fair comparison of the algorithms, the participants of the study had no knowledge of the gold standard results until after their results had been submitted. A few years later, the ImageCLEF[5,6] evaluation campaign introduced a challenge on medical image retrieval[7], based on experiences in the text retrieval domain where systematic evaluation had been performed since the 1960s[8]. About one decade ago, a broader interest in biomedical challenge organization arose with the first grand challenge that was organized in the scope of the international conference on Medical Image Computing and Computer Assisted Intervention (MICCAI) 2007[9]. Over time, research practice began to change, and the number of challenges organized annually has been increasing steadily (Fig. 1a), with currently about 28 biomedical image analysis challenges with a mean of 4 tasks conducted annually. Today, biomedical image analysis challenges are often published in prestigious journals (e.g[9–44].) and receive a huge amount of attention with hundreds of citations and thousands of views. Awarding the winner with a significant amount of prize money (up to €1 million on platforms like Kaggle[45]) is also becoming increasingly common.

This development was a great step forward, yet the increasing scientific impact[46,47] of challenges now puts huge responsibility on the shoulders of the challenge hosts that take care of the organization and design of such competitions. The performance of an algorithm on challenge data is essential, not only for the acceptance of a paper and its impact on the community, but also for the individuals' scientific careers, and the potential that algorithms can be translated into clinical practice. Given that this is so important, it is surprising that no commonly respected quality control processes for biomedical challenge design exist to date. Similar problems exist in other research communities, such as computer vision and machine learning.

In this paper, we present the first comprehensive evaluation of biomedical image analysis challenges based on 150 challenges conducted up until the end of 2016. It demonstrates the crucial nature of challenges for the field of biomedical image analysis, but also reveals major problems to be addressed: Reproduction, adequate interpretation, and cross-comparison of results are not possible in the majority of challenges, as only a fraction of the relevant information is reported and challenge design (e.g. a choice of metrics and methods for rank computation) is highly heterogeneous. Furthermore, the rank of an algorithm in a challenge is sensitive to a number of design choices, including the test data sets used for validation, the observer(s) who annotated the data and the metrics chosen for performance assessment, as well as the methods used for aggregating values.

## Results

**150 biomedical image analysis challenges**. Up until the end of 2016, 150 biomedical image analysis challenges that met our inclusion criteria (see Methods and Supplementary Table 1) were conducted with a total of 549 different image analysis tasks (see Fig. 1). 57% of these challenges (75% of all tasks) published their results in journals or conference proceedings. The information

used in this paper from the remaining challenges was acquired from websites. Most tasks were related to segmentation (70%) and classification (10%) and were organized within the context of the MICCAI conference (50%), and the IEEE International Symposium on Biomedical Imaging (ISBI) (following at 34%). The majority of the tasks dealt with 3D (including 3D + t) data (84%), and the most commonly applied imaging techniques were magnetic resonance imaging (MRI) (62%), computed tomography (CT) (40%) and microscopy (12%). The percentage of tasks that used in vivo, in silico, ex vivo, in vitro, post mortem, and phantom data was 85, 4, 3, 2, 2, and 1%, respectively (9%: N/A; 3%: combination of multiple types). The in vivo data was acquired from patients in clinical routine (60%), from patients under controlled conditions (9%), from animals (8%), from healthy human subjects (5%), or from humans under unknown (i.e. not reported) conditions (32%). While training data is typically provided by the challenge organizers (85% of all tasks), the number of training cases varies significantly across the tasks (median: 15; interquartile range (IQR): (7, 30); min: 1, max: 32,468). As with the training cases, the number of test cases varies across the tasks (median: 20; IQR: (12, 33); min: 1, max: 30,804). The median ratio of training cases to test cases was 0.75. The test data used differs considerably from the training data, not only in quantity but also in quality. For 73% of all tasks with human or hybrid reference generation, multiple observers have annotated the reference data. In these cases, an image was annotated by a median of 3 (IQR: (3, 4), max: 9) observers.

**Half of the relevant information is not reported**. We identified the relevant parameters that characterize a biomedical challenge following an ontological approach (see Methods). This yielded a total of 53 parameters corresponding to the categories challenge organization, participation conditions, mission of the challenge, study conditions, challenge data sets, assessment method, and challenge outcome (see Table 1). A biomedical challenge task reported a median of 62% (IQR: (51, 72%); min: 21%, max: 92%) of these parameters. 6% of the parameters were reported for all tasks and 43% of all parameters were reported for <50% of all tasks. The list of parameters which are generally not reported includes some that are crucial for interpretation of results. For example, 8% of all tasks providing an aggregated ranking across multiple metrics did not report the rank aggregation method they used (i.e. the method according to which the winner has been determined). Eighty five percent of the tasks did not give instructions on whether training data provided by challenge organizers may have been supplemented by other publicly available or private data, although the training data used is key to the success of any machine learning algorithm (see e.g[48].). In 66% of all tasks, there was no description on how the reference (i.e. gold standard) annotation was performed although the quality of annotation in the field of biomedical image analysis varies dependent on the user[49]. Forty five percent of tasks with multiple annotators did not describe how the annotations were aggregated. Also, the level of expertize of the observers that annotated the reference data was often (19%) not described. When analyzing the parameter coverage for different years, algorithm categories, publication venues, and platforms (see Supplementary Note 1), one main observation was that those challenges that were only represented on websites showed a substantial difference in quality of reporting when compared to those published in a journal or in conference proceedings. The supplementary material further shows how the reporting of individual parameters evolves over time.

**Large variability in challenge design**. In total, 97 different metrics have been used for performance assessment (three on

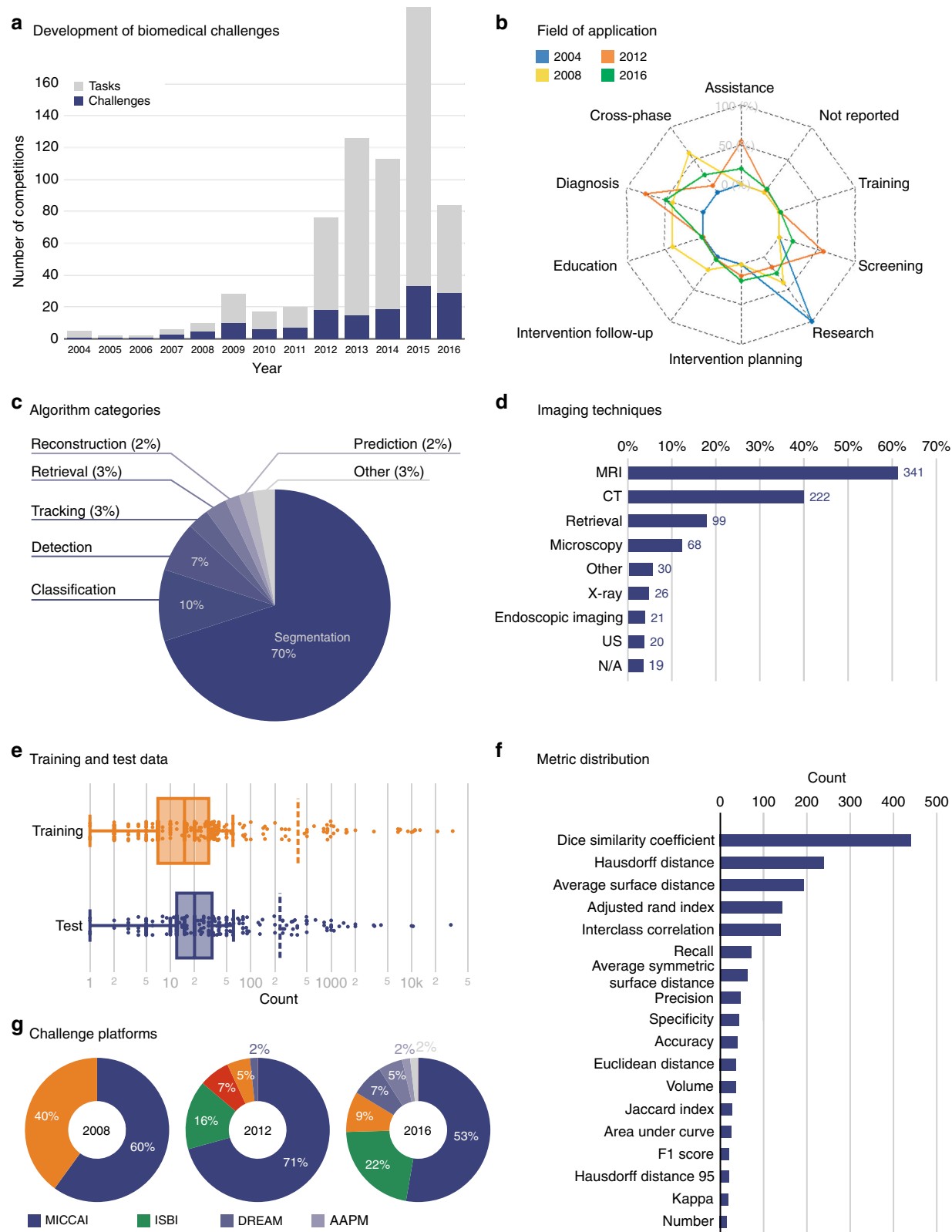

**Fig. 1** Overview of biomedical image analysis challenges. **a** Number of competitions (challenges and tasks) organized per year, **b** fields of application, **c** algorithm categories assessed in the challenges, **d** imaging techniques applied, **e** number of training and test cases used, **f** most commonly applied metrics for performance assessment used in at least 5 tasks, and **g** platforms (e.g. conferences) used to organize the challenges for the years 2008, 2012, and 2016

**Table 1 List of parameters that characterize a challenge**

| Parameter name | Coverage [%] | Parameter name | Coverage [%] |
|---|---|---|---|
| Challenge name[a] | 100 | Operator(s) | 7 |
| Challenge website[a] | 99 | Distribution of training and test cases [a] | 18 |
| Organizing institutions and contact person[a] | 97 | Category of training data generation method[a] | 89 |
| Life cycle type[a] | 100 | Number of training cases[a] | 89 |
| Challenge venue or platform | 99 | Characteristics of training cases[a] | 79 |
| Challenge schedule[a] | 81 | Annotation policy for training cases[a] | 34 |
| Ethical approval[a] | 32 | Annotator(s) of training cases[a] | 81 |
| Data usage agreement | 60 | Annotation aggregation method(s) for training cases[a] | 30 |
| Interaction level policy[a] | 62 | Category of test data generation method[a] | 87 |
| Organizer participation policy[a] | 6 | Number of test cases[a] | 77 |
| Training data policy[a] | 16 | Characteristics of test cases[a] | 77 |
| Pre-evaluation method | 5 | Annotation policy for test cases[a] | 34 |
| Evaluation software | 26 | Annotator(s) of test cases[a] | 78 |
| Submission format[a] | 91 | Annotation aggregation method(s) for test cases[a] | 34 |
| Submission instructions | 91 | Data pre-processing method(s) | 24 |
| Field(s) of application[a] | 97 | Potential sources of reference errors | 28 |
| Task category(ies)[a] | 100 | Metric(s)[a] | 96 |
| Target cohort* | 65 | Justification of metrics[a] | 23 |
| Algorithm target(s)[a] | 99 | Rank computation method[a] | 36 |
| Data origin[a] | 98 | Interaction level handling[a] | 44 |
| Assessment aim(s)[a] | 38 | Missing data handling[a] | 18 |
| Study cohort[a] | 88 | Uncertainty handling[a] | 7 |
| Context information[a] | 35 | Statistical test(s)[a] | 6 |
| Center(s)[a] | 44 | Information on participants | 88 |
| Imaging modality(ies)[a] | 99 | Results | 87 |
| Acquisition device(s) | 25 | Report document | 74 |
| Acquisition protocol(s) | 72 | | |

List of parameters that were identified as relevant when reporting a challenge along with the percentage of challenge tasks for which information on the parameter has been reported. Parameter definitions can be found in *Supplementary Table 2*.
[a]Parameters used for structured challenge submission for the MICCAI 2018 challenges

average per task). Metric design is very heterogeneous, particularly across comparable challenges, and justification for a particular metric is typically (77%) not provided. Roughly half of all metrics (51%) were only applied on a single task. Even in the main field of medical image segmentation, 34% of the 38 different metrics used were only applied once. The fact that different names may sometimes refer to the same metric was compensated for in these computations. Thirty nine percent of all tasks provided a final ranking of the participants and thus determined a challenge winner. Fifty seven percent of all tasks that provide a ranking do so on the basis of a single metric. In this case, either metric-based (aggregate, then rank; 76%) or case-based (rank per case, then aggregate; 1%) is typically performed (see Methods and Supplementary Discussion). Overall, 10 different methods for determining the final rank (last step in computation) of an algorithm based on multiple metrics were applied.

**Minor changes in metrics may make the last the first**. Besides the Dice Similarity Coefficient (DSC)[50], which was used in 92% of all 383 segmentation tasks (2015: 100%), the Hausdorff Distance (HD)[51,52] is the most commonly applied metric in segmentation tasks (47%). It was used either in its original formulation (42%) or as the 95% variant (HD95) (5%) (38%/8% in the 2015 segmentation challenges). We determined a single-metric ranking based on both versions for all 2015 segmentation challenges and found radical differences in the rankings as shown in Fig. 2a). In one case, the worst-performing algorithm according to the HD (10th place) was ranked first in a ranking based on the HD95.

**Different aggregation methods produce different winners**. One central result of most challenges is the final ranking they produce. Winners are considered state of the art and novel contributions

are then benchmarked according to them. The significant design choices related to the ranking scheme based on one or multiple metric(s) are as follows: whether to perform metric-based (aggregate, then rank) or case-based (rank, then aggregate) and whether to take the mean or the median. Statistical analysis with Kendall's tau (rank correlation coefficient[53]) using all segmentation challenges conducted in 2015 revealed that the test case aggregation method has a substantial effect on the final ranking, as shown in Fig. 2b, c). In some cases, almost all teams change their ranking position when the aggregation method is changed. According to bootstrapping experiments (Figs. 3 and 4), single-metric rankings are statistically highly significantly more robust when (1) the mean rather than the median is used for aggregation and (2) the ranking is performed after the aggregation.

**Different annotators produce different winners**. In most segmentation tasks (62%), it remains unclear how many observers annotated the reference data. Statistical analysis of the 2015 segmentation challenges, however, revealed that different observers may produce substantially different rankings, as illustrated in Fig. 2d. In experiments performed with all 2015 segmentation challenges that had used multiple observers for annotation (three tasks with two observers, one with five observers), different observers produced different rankings in 15, 46, and 62% of the 13 pairwise comparisons between observers, when using a single-metric ranking with mean aggregation based on the DSC, HD, and HD95, respectively. In these cases, the ranges of Kendall's tau were [0.78, 1], [−0.02, 1], and [0.07, 1], respectively.

**Removing one test case can change the winner**. Ideally, a challenge ranking should reflect the algorithms' performances for a task and thus be independent of the specific data sets used for

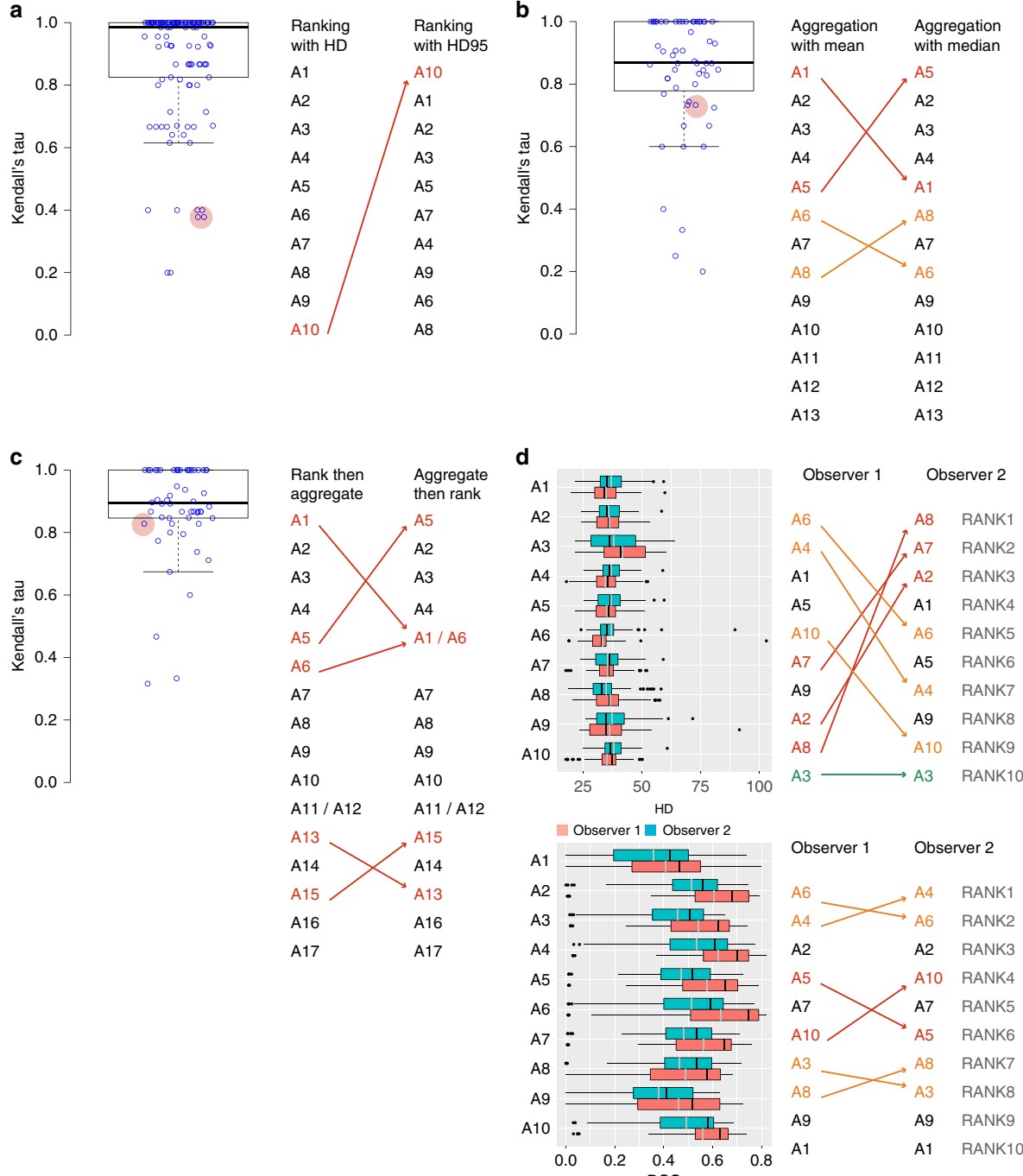

**Fig. 2** Robustness of rankings with respect to several challenge design choices. One data point corresponds to one segmentation task organized in 2015 ($n = 56$). The center line in the boxplots shows the median, the lower, and upper border of the box represent the first and third quartile. The whiskers extend to the lowest value still within 1.5 interquartile range (IQR) of the first quartile, and the highest value still within 1.5 IQR of the third quartile. **a** Ranking (metric-based) with the standard Hausdorff Distance (HD) vs. its 95% variant (HD95). **b** Mean vs. median in metric-based ranking based on the HD. **c** Case-based (rank per case, then aggregate with mean) vs. metric-based (aggregate with mean, then rank) ranking in single-metric ranking based on the HD. **d** Metric values per algorithm and rankings for reference annotations performed by two different observers. In the box plots (**a**–**c**), descriptive statistics for Kendall's tau, which quantifies differences between rankings (1: identical ranking; −1: inverse ranking), is shown. Key examples (red circles) illustrate that slight changes in challenge design may lead to the worst algorithm (A$_i$: Algorithm i) becoming the winner (**a**) or to almost all teams changing their ranking position (**d**). Even for relatively high values of Kendall's tau (**b**: tau = 0.74; **c**: tau = 0.85), critical changes in the ranking may occur

validation. However, a re-evaluation of all segmentation challenges conducted in 2015 revealed that rankings are highly sensitive to the test data applied (Fig. 4). According to bootstrapping experiments with the most commonly applied segmentation

metrics, the first rank is stable (the winner stays the winner) for 21, 11, and 9% of the tasks when generating a ranking based on the DSC, the HD or the 95% variant of the HD, respectively. For the most frequently used metric (DSC; 100% of all

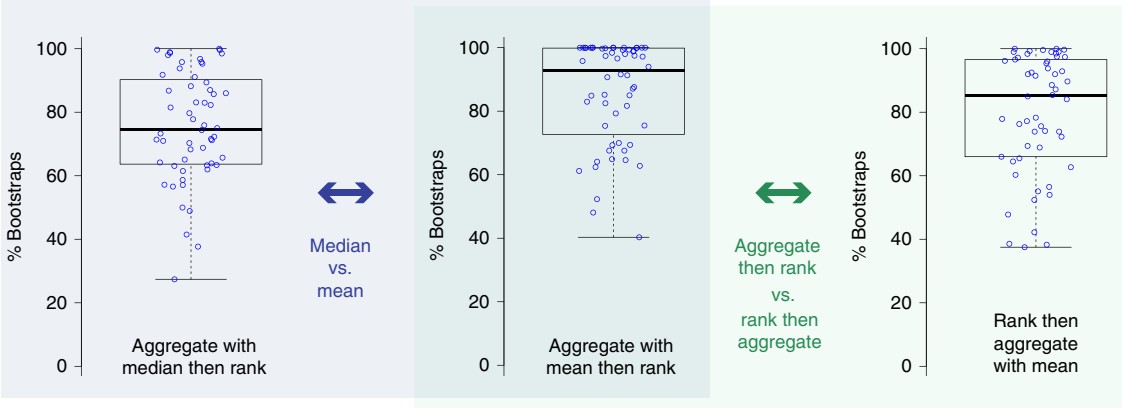

**Fig. 3** The ranking scheme is a deciding factor for the ranking robustness. The center line in the boxplots shows the median, the lower, and upper border of the box represent the first and third quartile. The whiskers extend to the lowest value still within 1.5 interquartile range (IQR) of the first quartile, and the highest value still within 1.5 IQR of the third quartile. According to bootstrapping experiments with 2015 segmentation challenge data, single-metric based rankings (those shown here are for the DSC) are significantly more robust when the mean rather than the median is used for aggregation (left) and when the ranking is performed after aggregation rather than before (right). One data point represents the robustness of one task, quantified by the percentage of simulations in bootstrapping experiments in which the winner remains the winner

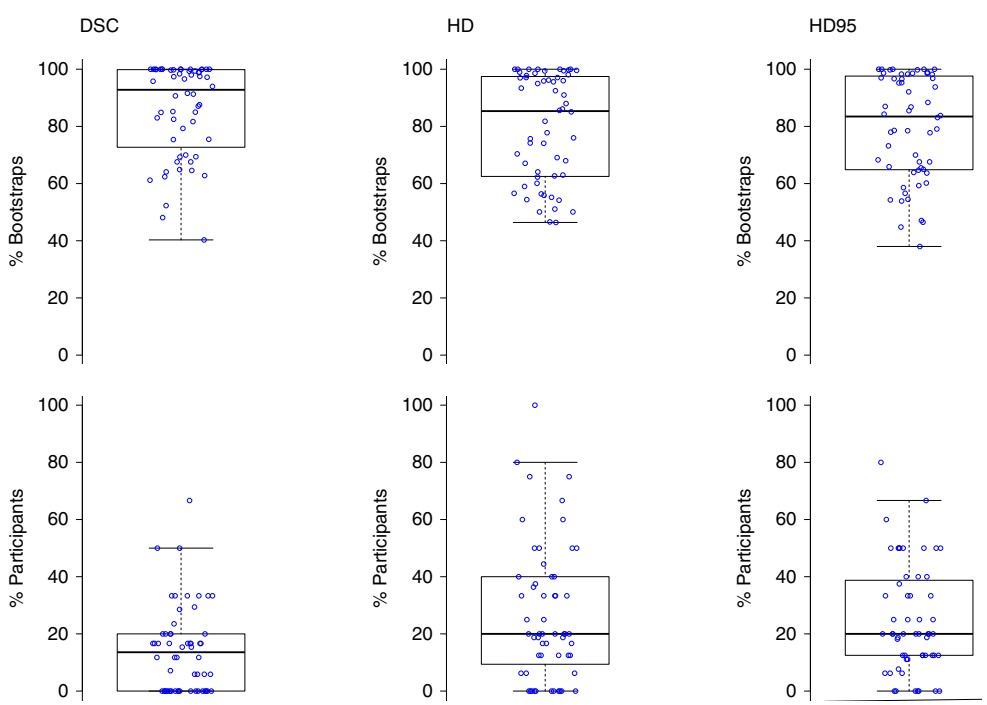

**Fig. 4** Robustness of rankings with respect to the data used. Robustness of rankings with respect to the data used when a single-metric ranking scheme based on whether the Dice Similarity Coefficient (DSC) (left), the Hausdorff Distance (HD) (middle) or the 95% variant of the HD (right) is applied. One data point corresponds to one segmentation task organized in 2015 ($n = 56$). The center line in the boxplots shows the median, the lower, and upper border of the box represent the first and third quartile. The whiskers extend to the lowest value still within 1.5 interquartile range (IQR) of the first quartile, and the highest value still within 1.5 IQR of the third quartile. Metric-based aggregation with mean was performed in all experiments. Top: percentage of simulations in bootstrapping experiments in which the winner (according to the respective metric) remains the winner. Bottom: percentage of other participating teams that were ranked first in the simulations

2015 segmentation challenges), a median of 15% and up to 100% of the other teams were ranked first in at least 1% of the bootstrap partitions. Even when leaving out only a single test case (and thus computing the ranking with one test case less), other teams than the winning team were ranked first in up to 16% of the cases. In one task, leaving a single test case out led to 67% of the teams other than the winning team ranking first.

**Lack of missing data handling allows for rank manipulation.** 82% of all tasks provide no information about how missing data is handled. While missing data handling is straightforward in case-based aggregation (the algorithms for which no results were submitted receive the last rank for that test case) it is more challenging in metric-based aggregation, especially when no worst possible value can be defined for a metric. For this reason,

several challenge designs simply ignore missing values when aggregating values. A re-evaluation of all 2015 segmentation challenges revealed that 25% of all 419 non-winning algorithms would have been ranked first if they had systematically just submitted the most plausible results (ranking scheme: aggregate DSC with mean, then rank). In 9% of the 56 tasks, every single participating team could have been ranked first if they had not submitted the poorest cases.

**Researchers request quality control**. Our experimental analysis of challenges was complemented by a questionnaire (see Methods). It was submitted by a total of 295 participants from 23 countries. 92% of participants agreed that biomedical challenge design should be improved in general, 87% of all participants would appreciate best practice guidelines, and 71% agreed that challenges should undergo more quality control. A variety of issues were identified for the categories data, annotation, evaluation, and documentation (cf. Figure 5). Many concerns involved the representativeness of the data, the quality of the (annotated) reference data, the choice of metrics and ranking schemes, and the lack of completeness and transparency in reporting challenge results. Details are provided in Supplementary Note 2 and Supplementary Methods.

**Complete reporting as a first step towards better practices**. Based on the findings of this study and the answers to the questionnaire, we have identified several best practice recommendations (see Supplementary Table 3) corresponding to the main problems in biomedical challenge design. The establishment of common standards and clear guidelines is currently hampered by open research questions that still need addressing. However, one primary practice that can be universally recommended is comprehensive reporting of the challenge design and results. Our practical and concrete recommendation is therefore to publish the complete challenge design before the challenge by instantiating the list of parameters proposed in this paper (Table 1, Supplementary Table 2). Three example instantiations are provided in Supplementary Table 2. The MICCAI 2018 satellite event team used the parameter list in the challenge proposal submission system to test its applicability. The submission system required a potential MICCAI 2018 challenge organizer to instantiate at least 90% of a reduced set of 40 parameters (cf. Table 1) that were regarded as essential for judging the quality of a challenge design proposal. The median percentage of parameters instantiated was 100% (min: 98%) (16 submitted challenges).

## Discussion

This paper shows that challenges play an increasingly important role in the field of biomedical image analysis, covering a huge range of problems, algorithm classes, and imaging modalities (Fig. 1). However, common practice related to challenge reporting is poor and does not allow for adequate interpretation and reproducibility of results (Table 1). Furthermore, challenge design is very heterogeneous and lacks common standards, although these are requested by the community (Table 1, Fig. 5). Finally, challenge rankings are sensitive to a range of challenge design parameters, such as the metric variant applied, the type of test case aggregation performed, and the observer annotating the data. The choice of metric and aggregation scheme has a significant influence on the ranking's stability (Figs. 2–4). Based on these findings and an international survey, we compiled a list of best practice recommendations and open research challenges (see Supplementary Table 3). The most universal recommendation is the instantiation of a list of 53 challenge parameters before

challenge execution to ensure fairness and transparency along with interpretability and reproducibility of results (Table 1).

One of the key implications of our findings is the discrepancy between the potential impact of challenges (e.g. finding solutions for the primary open problems in the field, identifying the best methods for classes of problems, establishing high-quality benchmarking data sets) and their current practical value. Our study shows that the specific (according to our questionnaire sometimes arbitrarily taken) challenge design choices (e.g. mean vs. median for metric value aggregation, number and expert level of data annotator(s), missing data handling etc.) have a crucial effect on the ranking. Hence, the challenge design – and not (only) the value of the methods competing in a challenge – may determine the attention that a particular algorithm will receive from the research community and from companies interested in translating biomedical research results.

As a consequence, one may wonder which conclusions may actually be drawn from a challenge. It seems only consequent to ask whether we should generally announce a winner at all. This question appears particularly interesting when considering that the competitive character of today's challenges may result in scientists investing valuable resources into fine-tuning their algorithms towards a specific challenge design instead of methodologically solving the underlying problem. For example, several challenges ignore missing values and it may be worth investing time into tuning a method such that results on difficult cases are simply not submitted and the overall mean/median performance is improved. A similar concern was recently raised in the related field of machine learning. Sculley et al[54]. postulate that emphasis on competitions to be won encourages parameter tuning on large machines at the expense of doing controlled studies to learn about an algorithm's strengths and weaknesses. Collaborative challenges without winners, which have been successfully applied in mathematics, for example[55,56], could potentially solve this issue to some extent but require a modular challenge design, which may not be straightforward to implement. Furthermore, the competition itself along with the opportunity to promote one's own methods are often key incentives for researchers to participate in a challenge, especially when they originate from a different (methodological) community, such as computer vision. The concept of combining competitive elements with collaborative elements, as pursued in the DREAM challenges[57], should be further investigated in this context.

Even if a specific challenge design resulted in a robust ranking (e.g. due to a huge number of test cases, appropriate metrics/ranking schemes and high-quality reference annotations), drawing broad conclusions from the challenge would not necessarily be straightforward[10]. A typically important question, for example, is whether a statistically significant difference in a metric value is clinically/biologically relevant. This may differ crucially from application to application. A related but increasingly relevant problem is the fact that it is often hard to understand which specific design choice of an algorithm actually makes this algorithm better than the competing algorithms. It is now well-known, for example, that the method for data augmentation (i.e. the way training cases are used to generate even more training data, e.g. by applying geometrical operations, such as mirroring and rotation, to both the input data and the reference annotations) often has a much bigger influence on the performance of a deep learning algorithm than the network architecture itself. Along these lines, Lipton and Steinhardt[58] point out that the way in which machine learning results are reported can sometimes be misleading, for example, by failing to identify the sources of empirical gains and through speculation disguised as explanation. The authors thus argue for a structured description not only of the challenge itself but also of the competing algorithms. Ideally,

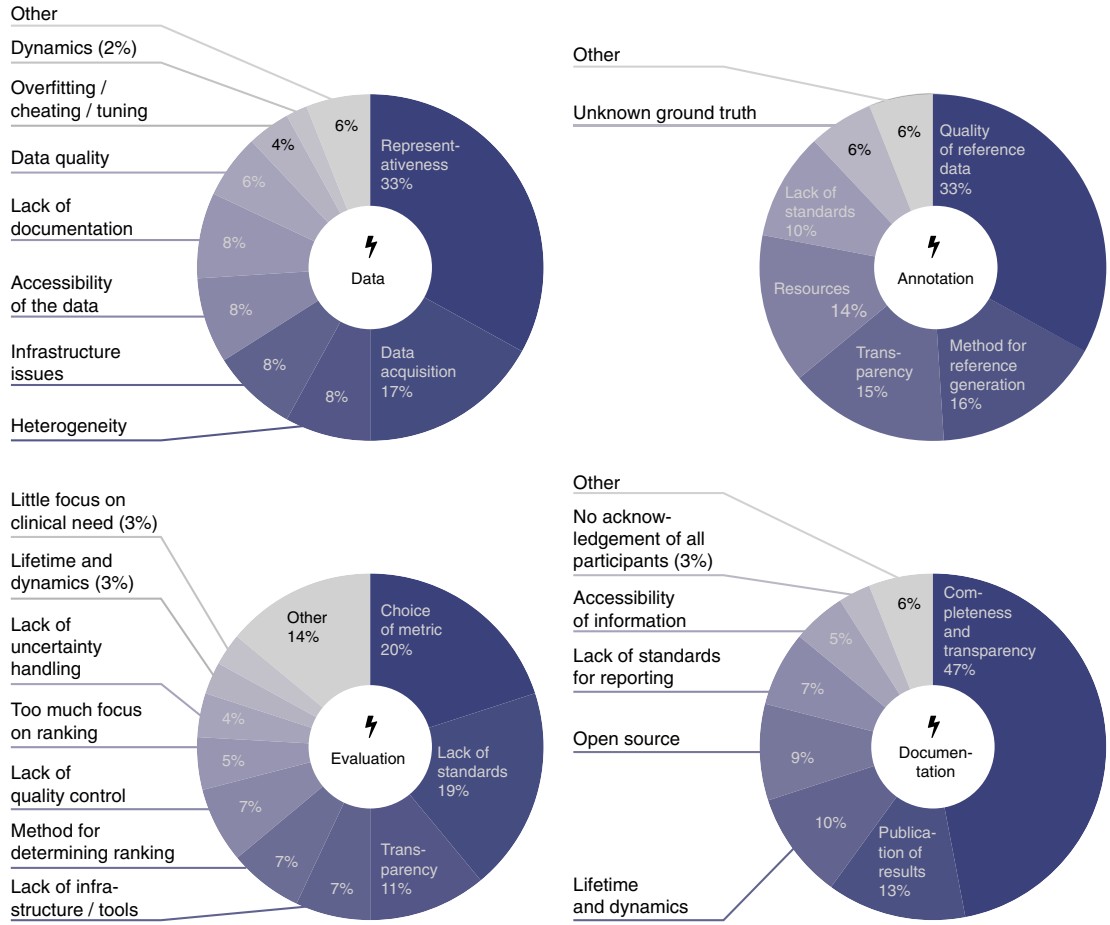

**Fig. 5** Main results of the international questionnaire on biomedical challenges. Issues raised by the participants were related to the challenge data, the data annotation, the evaluation (including choice of metrics and ranking schemes) and the documentation of challenge results

competing methods would be released open source (admittedly a potential problem for participants from industry), and a structured description of the method would be generated automatically from the source code. Due to the lack of common software frameworks and terminology, however, this is far from straightforward to implement at this stage.

An overarching question related to this paper is whether not only the control of the challenge design but also the selection of challenges should be encouraged. Today, the topics that are being pursued in the scope of challenges are not necessarily related to the actual grand challenges that the communities face. Instead, they are a result of who is willing and allowed to release their data and dedicate resources to organizing a competition. Given the fact that the pure existence of benchmarking data sets for a particular problem clearly leads to more people investing resources into the topic, mechanisms should be put in place to additionally channel the resources of the scientific community to the most important unsolved problems.

Overall, the demand for improvement along with the complexity of the problem raises the question of responsibility. The authors encourage the different stakeholders involved in challenge design, organization, and reporting to help overcome systemic hurdles.

Societies in the field of biomedical image processing should make strategic investments to increase challenge quality. One practical recommendation would be to establish the concept of challenge certification. Analogously to the way clinical studies can

be classified into categories reflecting the evidence level (e.g. case report vs. retrospective analysis vs. randomized double-blinded prospective study), challenges should be classified and certified according to criteria related to comprehensiveness of reporting, challenge design in the context of common practice as well as the data quantity and quality. Ideally, the certification would include a control process for the reference annotations. The authors believe that such a certification process should be handled by the societies related to the target domain rather than by platforms hosting challenges (such as Kaggle), which may lack the necessary medical/biological expertize to implement such a mechanism. Similarly, the authors see it as the role of the societies to release best practice recommendations for challenge organization in the different fields that require dedicated treatment.

In turn, platforms hosting challenges should perform a much more rigorous quality control. To improve challenge quality, for example, it should be made possible to give open feedback on the data and design of challenges (e.g. ability to report erroneous annotations). Furthermore, a more rigorous review of challenge proposals should be put in place by conferences. In a first attempt to establish a structured challenge review process, the organizers of this year's MICCAI used the parameter list presented in this paper as a basis for structured challenge proposal submission. While the instantiation of the list can be regarded as cumbersome, the authors believe that such a manner of quality control is essential to ensure reproducibility and interpretability of results. This initiative, however, can only be regarded as a first step, also

because control mechanisms to ensure that the proposed challenge designs will be implemented as suggested are resource-intensive and still lacking. Furthermore, the parameter list still lacks (external) instantiation from some domains, especially in the field of biological image analysis.

Funding organizations should dedicate resources for addressing the open research questions summarized in Supplementary Table 3. They should further identify open problems in the field of biomedical image analysis that should be tackled in the scope of either collaborative or competitive challenges and provide funding for the design, organization, and certification of these challenges. This is in contrast to common practice where funding is typically provided for solving specific problems.

Journal editors and reviewers should provide extrinsic motivation to raise challenge quality by establishing a rigorous review process. Several high-impact journals have already taken important measures to ensure reproducibility of results in general. These should be complemented by concepts for quality control regarding comprehensiveness of reporting, generation of reference annotations and choice of metrics and ranking schemes. Furthermore, journal editors are encouraged to work with the respective societies to establish best practice recommendations for all the different subfields of a domain, e.g. by initiating special issues dedicated to best practices in validation and challenge design.

Organizers of challenges are highly encouraged to follow the recommendations summarized in this paper and to contribute to the establishment of further guidelines dedicated to specific subfields of biomedical image analysis. They should put a particular focus on the generation of high-quality reference data and the development and deployment of an infrastructure that prevents cheating and overfitting to the challenge data.

Finally, scientists are encouraged to dedicate their resources to the open research questions identified (Supplementary Table 3) and to contribute their data and resources to the establishment of high-quality benchmarking data sets.

While this paper concentrates on the field of biomedical image analysis challenges, its impact can be expected to go beyond this field. Importantly, many findings of this paper apply not only to challenges but to the topic of validation in general. It may be expected that more effort is typically invested when designing and executing challenges (which, by nature, have a high level of visibility and go hand in hand with publication of the data) compared to the effort invested in performing in-house studies dedicated to validation of an individual algorithm. Therefore, concerns involving the meaningfulness of research results in general may be raised. This may also hold true for other research fields, both inside and outside the life sciences, as supported by related literature[59–63].

Clearly, it will not be possible to solve all the issues mentioned in a single large step. The challenge framework proposed could be a good environment in which to start improving common practice of benchmarking. Implementing a (possibly domain-specific) checklist of parameters to be instantiated in order to describe the data used in a challenge can safely be recommended across scientific disciplines. In the long run, this could encourage further improvements in the documentation of the algorithms themselves.

In conclusion, challenges are an essential component in the field of biomedical image analysis, but major research challenges and systemic hurdles need to be overcome to fully exploit their potential to move the field forward.

## Methods

**Definitions**. We use the following terms throughout the paper:

Challenge: open competition on a dedicated scientific problem in the field of biomedical image analysis. A challenge is typically organized by a consortium that issues a dedicated call for participation. A challenge may deal with multiple different tasks for which separate assessment results are provided. For example, a challenge may target the problem of segmentation of human organs in computed tomography (CT) images. It may include several tasks corresponding to the different organs of interest.

Task: subproblem to be solved in the scope of a challenge for which a dedicated ranking/leaderboard is provided (if any). The assessment method (e.g. metric(s) applied) may vary across different tasks of a challenge.

Case: data set for which the algorithm(s) of interest produce one result in either the training phase (if any) or the test phase. It must include one or multiple images of a biomedical imaging modality (e.g. a CT and a magnetic resonance imaging (MRI) image of the same structure) and typically comprises a gold standard annotation (usually required for test cases).

Metric: a measure (not necessarily metric in the strict mathematical sense) used to compute the performance of a given algorithm for a given case, typically based on the known correct answer. Often metrics are normalized to yield values in the interval from 0 (worst performance) to 1 (best performance).

Metric-based vs. case-based aggregation: to rank an algorithm participating in a challenge based on the performance on a set of test cases according to one or multiple metrics, it is necessary to aggregate values to derive a final rank. In single-metric rankings, we distinguish the following two categories, which cover most ranking schemes applied. Metric-based aggregation begins with aggregating metric values over all test cases (e.g. with the mean or median). Next, a rank for each algorithm is computed. In contrast, case-based aggregation begins with computing a rank for each test case for each algorithm. The final rank is determined by aggregating test case ranks (see Supplementary Discussion for more details).

**Inclusion criteria**. Inclusion criteria for "Experiment: Comprehensive reporting": Our aim was to capture all biomedical image analysis challenges that have been conducted up to 2016. We did not include 2017 challenges as our focus is on information provided in scientific papers, which may have a delay of more than a year to be published after challenge execution. To acquire the data, we analyzed the websites hosting/representing biomedical image analysis challenges, namely grand-challenge.org, dreamchallenges.org, and kaggle.com as well as websites of main conferences in the field of biomedical image analysis, namely Medical Image Computing and Computer Assisted Intervention (MICCAI), International Symposium on Biomedical Imaging (ISBI), International Society for Optics and Photonics (SPIE) Medical Imaging, Cross Language Evaluation Forum (CLEF), International Conference on Pattern Recognition (ICPR), The American Association of Physicists in Medicine (AAPM), the Single Molecule Localization Microscopy Symposium (SMLMS) and the BioImage Informatics Conference (BII). This yielded a list of 150 challenges with 549 tasks.

Inclusion criteria for "Experiment: Sensitivity of challenge ranking": all organizers of 2015 segmentation challenges ($n = 14$) were asked to provide the challenge results (per algorithm and test case) and (re-)compute a defined set of common performance measures, including the Dice Similarity Coefficient (DSC) and the Hausdorff Distance (HD) in the original version[51] and the the 95% variant (HD95)[52]. While the DSC was used in the original design of all 2015 challenges, the HD/HD95 was not always applied. In all, 13 challenges were able to provide the measures as requested. These challenges are composed of 124 tasks in total. The specific inclusion criteria on challenge and task level are provided in Tables 2 and 3.

**Challenge parameter list**. One key purpose of this paper was to develop a list of parameters that can be instantiated for describing the design and results of a challenge in a comprehensive manner, thus facilitating interpretability and reproducibility of results. To this end, the following procedure was followed:

Initialization: the parameters for describing reference-based validation studies presented in ref. [64] served as an initial set.

Adding challenge-specific parameters: during analysis of challenge websites and papers, the initial list was complemented such that the information available on a challenge could be comprehensively formalized.

Refinement based on challenge capturing: a tool was designed to formalize existing challenges with the current parameter list. During this process, the list was further refined.

Refinement with international questionnaire: Finally, a questionnaire was designed and sent to all co-authors to finalize the list. All participants were asked to comment on the name, the description, the importance and possible instantiations of each parameter. Adding further parameters was also allowed.

Finalization with ontological modeling: based on the final list, an ontology for describing biomedical image analysis challenges was developed. The latter was used for structured submission of MICCAI 2018 biomedical challenges.

**Statistical methods**. To quantify the robustness of a ranking, the following statistical methods were used:

Kendall's tau analysis: to quantify the agreement of two rankings (e.g. for two different aggregation methods or two different metric variants), Kendall's tau (also named Kendall's rank correlation or simply tau)[53] was determined as recommended in ref. [65]. Tau was designed to be independent of the number of

**Table 2 Inclusion criteria on challenge level**

| # | Criterion | Number of affected tasks/ challenges |
|---|-----------|------------------|
| 1 | If a challenge task has on- and off-site part, the results of the part with the most participating algorithms are used. | 1/1 |
| 2 | If multiple reference annotations are provided for a challenge task and no merged annotation is available, the results derived from the second annotator are used. In one challenge, the first annotator produced radically different annotations from all other observers. This is why we used the second observer of all challenges. | 2/2 |
| 3 | If multiple reference annotations are provided for a challenge task and a merged annotation is available, the results derived from the merged annotation are used. | 1/1 |
| 4 | If an algorithm produced invalid values for a metric in all test cases of a challenge task, this algorithm is omitted in the ranking | 1/1 |

**Table 3 Inclusion criteria on task level**

| # | Criterion | Number of excluded tasks |
|---|-----------|--------------|
| 1 | Number of algorithms ≥3 | 42 |
| 2 | Number of test cases >1 (for bootstrapping and cross-validation approaches) | 25 |
| 3 | No explicit argumentation against the usage of Hausdorff Distance as metric | 1 |

entities ranked and may take values between 1 (perfect agreement, i.e. same ranking) and −1 (reverse ranking).

Bootstrapping: for analysis of the variability of a ranking scheme (e.g. as a function of the metric applied), the following bootstrap approach was chosen: for a given task, the original ranking based on all test cases and a given ranking scheme as well as the winning algorithm according to this ranking scheme was determined. In all analyses, 1000 bootstrap samples were drawn from the data sets and the ranking scheme was applied to each bootstrap sample. It should be noted that on average, 63.2% of distinct data sets are retained in a bootstrap sample. For summary of the ranking scheme variability, the frequency of rank 1 in the bootstrap samples for the original winner (the winner remains the winner) as well as the proportion of algorithms that achieved rank 1 in the bootstraps but were not winning in the original ranking was determined. Competitions with multiple winners according to the original ranking were not included in the analysis (this occurred in just one task). For comparison of the stability of different ranking schemes, the same bootstrap samples were evaluated with different ranking schemes and a paired comparison between the proportion of the winner remaining the winner was performed by Wilcoxon signed rank test. Results were considered significant for $p < 0.05$.

Leave-one-out: for a given task, the original ranking based on all test cases and a given ranking scheme and the winning algorithm according to this ranking scheme was determined. The number of data sets was reduced by one and the ranking scheme was applied to this subset of data sets. The same summary measures as for the bootstrapping approach were determined.

Note that we did not rely on results of statistical testing approaches to quantify the stability of a given ranking scheme. The reasons for this decision were the following:

(a) The number of data sets varies widely between different tasks and due to correlation of power and sample size, results of statistical tests between different tasks are not comparable by design.

(b) If one were to use statistical testing, the appropriate approach would be to use a mixed model with a random factor for the data set and test the global hypothesis that all algorithms produce the same result, followed by post-hoc all pairwise comparisons. Pairwise comparisons would have to be adjusted for multiplicity and adjustment depends on the number of algorithms in the task. Again, results of statistical testing between different tasks are not comparable by design.

(c) We have evaluated the concordance of the bootstrap analysis for variability of ranking with a statistical testing approach and found examples where there was a highly significant difference between the winner and the second, but bootstrap analysis showed that ranking was very variable, and vice versa.

Boxplots with and without dots were produced to visualize results. In all boxplots, the boldfaced black line represents the median while the box represents the first and third quartile. The upper whisker extends to the largest observation ≤median + 1.5 IQR, and likewise the lower whisker to the smallest observation

≥median −1.5 IQR. In horizontal boxplots, the mean is shown in addition as boldfaced gray line.

All statistical analyses were performed with R version 3.4.3 (The R Foundation for Statistical Computing 2017). The figures were produced with Excel, R, Plotly (Python), and Adobe Illustrator 2017.

**Experiment: comprehensive reporting**. The key research questions corresponding to the comprehensive challenge analysis were:

RQ1: What is the role of challenges for the field of biomedical image analysis (e.g. How many challenges conducted to date? In which fields? For which algorithm categories? Based on which modalities?)

RQ2: What is common practice related to challenge design (e.g. choice of metric(s) and ranking methods, number of training/test images, annotation practice etc.)? Are there common standards?

RQ3: Does common practice related to challenge reporting allow for reproducibility and adequate interpretation of results?

To answer these questions, a tool for instantiating the challenge parameter list (Supplementary Table 2) introduced in the Methods section "Challenge parameter list" was used by some of the authors (engineers and a medical student) to formalize all challenges that met our inclusion criteria as follows: (1) Initially, each challenge was independently formalized by two different observers. (2) The formalization results were automatically compared. In ambiguous cases, when the observers could not agree on the instantiation of a parameter - a third observer was consulted, and a decision was made. When refinements to the parameter list were made, the process was repeated for missing values. Based on the formalized challenge data set, a descriptive statistical analysis was performed to characterize common practice related to challenge design and reporting.

**Experiment: sensitivity of challenge ranking**. The primary research questions corresponding to the experiments on challenge rankings were:

RQ4: How robust are challenge rankings? What is the effect of

(a) the specific test cases used?

(b) the specific metric variant(s) applied?

(c) the rank aggregation method chosen (e.g. aggregation of metric values with the mean vs median)?

(d) the observer who generated the reference annotation?

RQ5: Does the robustness of challenge rankings vary with different (commonly applied) metrics and ranking schemes?

RQ6: Can common practice on missing data handling be exploited to manipulate rankings?

As published data on challenges typically do not include metric results for individual data sets, we addressed these open research questions by approaching all organizers of segmentation challenges conducted in 2015 and asking them to provide detailed performance data on their tasks (124 in total). Note in this context that segmentation is by far the most important algorithm category (70% of all biomedical image analysis challenges) as detailed in the Results section. Our comprehensive challenge analysis further revealed single-metric ranking with mean and metric-based aggregation as the most frequently used ranking scheme. This is hence considered the default ranking scheme in this paper.

Our analysis further identified the DSC (92%) and the HD (47%) as the most commonly used segmentation metrics. The latter can either be applied in the original version (42%) or the 95% variant (HD95) (5%).

To be able to investigate the sensitivity of rankings with respect to several challenge design choices, the 2015 segmentation challenge organizers were asked to provide the assessment data (results for DSC, HD, and HD95) on a per data set basis for their challenge. The research questions RQ4-6 were then addressed with the following experiments:

RQ4: for all 56 segmentation tasks that met our inclusion criteria, we generated single-metric rankings with the default ranking scheme based on the DSC and the HD. We then used Kendall's tau to investigate the effect of changing (1) the metric

variant (HD vs HD95), (2) the aggregation operator (mean vs median), (3) the aggregation category (metric-based vs case-based), and (4) the observer (in case multiple annotations were available). Note in this context that we focused on single-metric rankings in order to perform a statistical analysis that enables a valid comparison across challenges.

RQ5: To quantify the robustness of rankings as a function of the metric, we generated single-metric rankings with the default ranking scheme based on the DSC, the HD, and the HD95. We then applied bootstrapping and leave-one-out analysis to quantify ranking robustness as detailed in Statistical Methods. Analogously, we compared the robustness of rankings for different aggregation methods (metric-based vs case-based) and aggregation operators (mean vs median).

RQ6: 82% of all biomedical image analysis tasks (see Results) do not report any information on missing values when determining a challenge ranking. In metric-based ranking (although not reported), it is common to simply ignore missing values. To investigate whether this common practice may be exploited by challenge participants to manipulate rankings, we performed the following analysis: For each algorithm and each task of each 2015 segmentation challenge that met our inclusion criteria, we determined the default ranking and artificially removed those test set results whose DSC was below a threshold of $t = 0.5$. Note that it can be assumed that these cases could have been relatively easily identified by visual inspection without comparing them to the reference annotations. We then compared the new ranking position of the algorithm with the position in the original (default) ranking.

**International survey.** As a basis for deriving best practice recommendations related to challenge design and organization, we designed a questionnaire (see Supplementary Methods) to gather known potential issues. It was distributed to colleagues of all co-authors, the challenges chairs of the past three MICCAI conferences as well as to the following mailing lists: ImageWorld, the mailing lists of the MICCAI society, the international society for computer aided surgery (ISCAS), the UK Euro-BioImaging project and the conferences Medical Image Understanding and Analysis (MIUA) and Bildverarbeitung für die Medizin (BVM). The link to the questionnaire was further published on grand-challenge.org.

## Data availability

Four data sets were generated and analyzed during the current study: DS1: captured biomedical challenges from publicly available sources (2004–2016). DS2: 2015 segmentation challenges results provided by challenge organizers. DS3: Individual responses to survey "Towards next-generation biomedical challenges". DS4: Individual responses to survey regarding refinement of parameter list. DS1 is available from Zenodo[66]. DS2 is not publicly available as it contains information that could compromise challenge participants' privacy or consent. DS3 and DS4 are available from the corresponding author L.M.-H. upon reasonable request. A reporting summary for this article is available as a Supplementary Information file.

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

## Acknowledgements

We thank all organizers of the 2015 segmentation challenges who are not co-authoring this paper (a list is provided as Supplementary Note 3) and all participants of the international questionnaire (a list is provided as Supplementary Note 4). We further thank Angelika Laha, Diana Mindroc-Filimon, Bünyamin Pekdemir, and Jenshika Yoganathan (DKFZ, Germany) for helping with the comprehensive challenge capturing.

Many thanks also go to Janina Dunning and Stefanie Strzysch (DKFZ, Germany) for their support of the project. Finally, we acknowledge support from the European Research Council (ERC) (ERC starting grant COMBIOSCOPY under the New Horizon Framework Programme grant agreement ERC-2015-StG-37960 as well as Seventh Framework Programme (FP7/2007-2013) under grant agreement no 318068 (VISCERAL)), the German Research Foundation (DFG) (grant MA 6340/10-1 and grant MA 6340/12-1), the Ministry of Science and Technology, Taiwan (MOST 106-3114-8-011-002, 106-2622-8-011-001-TE2, and 105-2221-E-011-121-MY2), the US National Institute of Health (NIH) (grants R01-NS070906, RG-1507-05243, and R01-EB017230 (NIBIB)), the Australian Research Council (DP140102794 and FT110100623), the Swiss National Science Foundation (grant 205321_157207), the Czech Science Foundation (grant P302/12/G157), the Czech Ministry of Education, Youth and Sports (grant LTC17016 in the frame of EU COST NEUBIAS project), the Engineering and Physical Sciences Research Council (EPSRC) (MedIAN UK Network (EP/N026993/1) and EP/P012841/1), the Wellcome Trust (NS/A000050/1), the Canadian Natural Science and Engineering Research Council (RGPIN-2015-05471), the UK Medical Research Council (MR/P015476/1), and the Heidelberg Collaboratory for Image Processing (HCI) including matching funds from the industry partners of the HCI.

## Author contributions

L.M.-H. initiated and designed the study. M.E., M.S., A.R., P.S., S.O., C.F., K.M., W.N., A. F.F., D.S., P.J., and L.M.-H. created the challenge parameter list with additional input from all co-authors. M.E., M.S., A.R., P.S., S.O., and P.M.F. acquired the publicly available data of all biomedical challenges conducted up to 2016. A.K.-S., C.S., M.E., A.R., S.O., and L.M.-H. designed and implemented the statistical analysis. F.v.d.S., A.C., G.C.S., B.H.M., S.S., B.A.L., K.S., O.M., G.Z., H.B., A.A.T., C.-W.W., A.P.B., and P.F.N. contributed data from 2015 segmentation challenges. A.R., M.E., and L.M.-H. designed the international questionnaire with additional input from all co-authors and analyzed the results. T.A., K.H. and B.H.M. provided background on research practices from other fields (computer vision and machine learning). M.-A.W. guided the project from a radiological perspective. K.H. headed the literature research and organization. L.M.-H. wrote the manuscript with substantial input from K.M.-H., H.M., M.K., A.P.B., A.H., B.v. G., N.R., and B.A.L, and feedback from all co-authors.

## Additional information

**Competing interests:** Henning Müller is on the advisory board of "Zebra Medical Vision". Danail Stoyanov is a paid part-time member of Touch Surgery, Kinosis Ltd. The remaining authors declare no competing interests.

Lena Maier-Hein [1], Matthias Eisenmann [1], Annika Reinke[1], Sinan Onogur[1], Marko Stankovic[1], Patrick Scholz[1], Tal Arbel[2], Hrvoje Bogunovic[3], Andrew P. Bradley[4], Aaron Carass [5], Carolin Feldmann[1], Alejandro F. Frangi [6],

Peter M. Full[1], Bram van Ginneken[7], Allan Hanbury[8,9], Katrin Honauer[10], Michal Kozubek[11], Bennett A. Landman[12], Keno März[1], Oskar Maier[13], Klaus Maier-Hein[14], Bjoern H. Menze[15], Henning Müller[16], Peter F. Neher[14], Wiro Niessen[17], Nasir Rajpoot[18], Gregory C. Sharp[19], Korsuk Sirinukunwattana[20], Stefanie Speidel[21], Christian Stock[22], Danail Stoyanov[23], Abdel Aziz Taha[24], Fons van der Sommen[25], Ching-Wei Wang[26], Marc-André Weber[27], Guoyan Zheng[28], Pierre Jannin[29] & Annette Kopp-Schneider[30]

[1]Division of Computer Assisted Medical Interventions (CAMI), German Cancer Research Center (DKFZ), 69120 Heidelberg, Germany. [2]Centre for Intelligent Machines, McGill University, Montreal, QC H3A0G4, Canada. [3]Christian Doppler Laboratory for Ophthalmic Image Analysis, Department of Ophthalmology, Medical University Vienna, 1090 Vienna, Austria. [4]Science and Engineering Faculty, Queensland University of Technology, Brisbane, QLD 4001, Australia. [5]Department of Electrical and Computer Engineering, Department of Computer Science, Johns Hopkins University, Baltimore, MD 21218, USA. [6]CISTIB - Center for Computational Imaging & Simulation Technologies in Biomedicine, The University of Leeds, Leeds, Yorkshire LS2 9JT, UK. [7]Department of Radiology and Nuclear Medicine, Medical Image Analysis, Radboud University Center, 6525 GA Nijmegen, The Netherlands. [8]Institute of Information Systems Engineering, TU Wien, 1040 Vienna, Austria. [9]Complexity Science Hub Vienna, 1080 Vienna, Austria. [10]Heidelberg Collaboratory for Image Processing (HCI), Heidelberg University, 69120 Heidelberg, Germany. [11]Centre for Biomedical Image Analysis, Masaryk University, 60200 Brno, Czech Republic. [12]Electrical Engineering, Vanderbilt University, Nashville, TN 37235-1679, USA. [13]Institute of Medical Informatics, Universität zu Lübeck, 23562 Lübeck, Germany. [14]Division of Medical Image Computing (MIC), German Cancer Research Center (DKFZ), 69120 Heidelberg, Germany. [15]Institute for Advanced Studies, Department of Informatics, Technical University of Munich, 80333 Munich, Germany. [16]Information System Institute, HES-SO, Sierre 3960, Switzerland. [17]Departments of Radiology, Nuclear Medicine and Medical Informatics, Erasmus MC, 3015 GD Rotterdam, The Netherlands. [18]Department of Computer Science, University of Warwick, Coventry CV4 7AL, UK. [19]Department of Radiation Oncology, Massachusetts General Hospital, Boston, MA 02114, USA. [20]Institute of Biomedical Engineering, University of Oxford, Oxford OX3 7DQ, UK. [21]Division of Translational Surgical Oncology (TCO), National Center for Tumor Diseases Dresden, 01307 Dresden, Germany. [22]Division of Clinical Epidemiology and Aging Research, German Cancer Research Center (DKFZ), 69120 Heidelberg, Germany. [23]Centre for Medical Image Computing (CMIC) & Department of Computer Science, University College London, London W1W 7TS, UK. [24]Data Science Studio, Research Studios Austria FG, 1090 Vienna, Austria. [25]Department of Electrical Engineering, Eindhoven University of Technology, 5600 MB Eindhoven, The Netherlands. [26]AIExplore, NTUST Center of Computer Vision and Medical Imaging, Graduate Institute of Biomedical Engineering, National Taiwan University of Science and Technology, Taipei 106, Taiwan. [27]Institute of Diagnostic and Interventional Radiology, University Medical Center Rostock, 18051 Rostock, Germany. [28]Institute for Surgical Technology and Biomechanics, University of Bern, Bern 3014, Switzerland. [29]Univ Rennes, Inserm, LTSI (Laboratoire Traitement du Signal et de l'Image) - UMR_S 1099, Rennes 35043 Cedex, France. [30]Division of Biostatistics, German Cancer Research Center (DKFZ), 69120 Heidelberg, Germany. These authors contributed equally: Lena Maier-Hein, Matthias Eisenmann. These authors jointly supervised this work: Pierre Jannin, Annette Kopp-Schneider.

