## [Peer Review File · Nature Communications]

REVIEWERS' COMMENTS:

Reviewer #2 (Remarks to the Author):

We have carefully studied the detailed response of the authors to our report as well as the corresponding changes in the manuscript and supplements. It is with great pleasure, and appreciation towards the authors, to report that the authors have implemented nearly all of our comments. For the one or two unimplemented ones, we accept their clear explanations. We have thus no further questions that we would see as a roadblock to accept the manuscript for publication.

We would only ask the authors to consider to tone down the word "controlled" in the sentence "An overarching question related to this paper is whether not only the challenge design but also the selection of challenges should be controlled." [page 16, 1st sentence of the 3rd paragraph]. We would propose something softer such as "desired" or "encouraged".... We do not think it is healthy for the field to attempt (which is what the word "controlled" suggests to us) to prevent anyone from organizing his own challenge "their own way", avoiding the recommendation of this manuscript. However, we absolutely agree with the authors that such a controlling mechanism should be implemented on the side of societies (and indeed not platforms) when deciding to provide the auspices or certification to a challenge.

BTW: There is probably a broken reference "Sculley et al. [ADD]" in the first paragraph on page 16.

Reviewer #3 (Remarks to the Author):

This is the revised version of the manuscript "Is the winner really the best? A critical analysis of common research practice in biomedical image analysis competitions" by Maier-Hein et al. which I had the opportunity to review previously.

As a first comment, I would like to acknowledge the work of the authors to revise the manuscript according to the referees' comments and suggestions, and for the new data and insights that do strengthen the work and the message.

That said, I still have some points of concern and discussion:

1. even though the authors have made an excellent "diagnostics" and indeed unearthed many problems associated with challenges, I feel (and fear) that the "remedy" part is rather weak and sounds more like wishful thinking than an effective and strong action plan. I do appreciate that the MICCAI board has taken some measures to improve on the challenges organisation, but there is a high risk this type of action remains isolated. This is why I believe this manuscript should be published by the different societies that are stakeholders of challenges in the form of a position paper simultaneously published in their journals or magazines, to give a strong support to the conclusions of this study.
2. I would have also appreciated that the authors go more into strong recommendations, in

particular concerning the metrics and the aggregation method(s). They rightly spot that the order of data aggregation and the metrics can influence the scoring. I would welcome a recommendation on which metrics to use and which ones to ban, for example why and when use the Hausdorff distance (HD) or the 95% variant (HD95), what are the advantages/caveats of the Jaccard index or why one should never use a given metrics. The same would be necessary for the aggregation methods.

3. In this context where the authors would indeed make recommendations rather than observations, I would suggest to drop the catchy (in the sense of sensationalistic) part of the title ("Is the winner really the best?") and spell out the purpose of the paper in the title "A critical analysis of ... and recommendations".

We would like to thank the reviewers and the editors again for all the effort they invested to help us improve the paper. We have revised the paper to incorporate the reviewers' second round of feedback. The most important changes are included below:

1. Title: We believe that an attention-grabbing and readily memorable title is important for an article of this nature. Nevertheless, we changed the title that may be perceived as less sensational version:

Why rankings of biomedical image analysis competitions should be interpreted with care

Alternatively, we suggest:

Challenging the criteria for nominating the winner in biomedical image analysis competitions

2. Supplementary Material on ranking schemes: In response to Reviewer 3's comment, we have generated a document that presents and compares three different methods for computing ranks: (1) Metric-based aggregation, the most widely used method, (2) case-based aggregation, the 2nd most commonly used ranking scheme, and (3) significance ranking, a scheme that the authors consider a promising alternative when requiring straightforward missing value handling and the natural assignment of identical ranks to algorithms that show only marginal differences.

Responses to all comments can be found below:

Reviewer 2 was generally satisfied with the revision ("It is with great pleasure, and appreciation towards the authors, to report that the

authors have implemented nearly all of our comments. For the one or two unimplemented ones, we accept their clear explanations.”) and only had two remaining minor comments:

R2-C1: We would only ask the authors to consider to tone down the word "controlled" in the sentence "An overarching question related to this paper is whether not only the challenge design but also the selection of challenges should be controlled." [page 16, 1st sentence of the 3rd paragraph]. We would propose something softer such as "desired" or "encouraged".... We do not think it is healthy for the field to attempt (which is what the word "controlled" suggests to us) to prevent anyone from organizing his own challenge "their own way", avoiding the recommendation of this manuscript. However, we absolutely agree with the authors that such a controlling mechanism should be implemented on the side of societies (and indeed not platforms) when deciding to provide the auspices or certification to a challenge.

Response: We have changed the sentence to: “An overarching question related to this paper is whether or not only the control of the challenge design but also the selection of challenges should be encouraged.”

R2-C2: BTW: There is probably a broken reference "Sculley et al. [ADD]" in the first paragraph on page 16.

Response: We have updated all references according to Nature Communications formatting guidelines.

Reviewer 3 acknowledged “the work of the authors to revise the manuscript according to the referees' comments and suggestions, and for the new data and insights that do strengthen the work and the message” and had three remaining comments:

R3-C1: even though the authors have made an excellent “diagnostics” and indeed unearthed many problems associated with challenges, I feel (and fear) that the “remedy” part is rather weak and sounds more like wishful thinking than an effective and strong action plan. I do appreciate that the MICCAI board has taken some measures to improve on the challenges organization, but there is a high risk this type of action remains isolated. This is why I believe this manuscript should be published by the different societies that are stakeholders of challenges in the form of a position paper simultaneously published in their journals or magazines, to give a strong support to the conclusions of this study.

Response:

- RE “remedy part weak”: We agree that the focus of the work was on the revealing of problems related to current challenge design. As an initial step towards better practices, we have compiled two pages of universal best practice recommendations including suggestions for further research. We believe, however, that more detailed recommendations are

out of scope for the present paper because many recommendations (e.g. choice of metrics) depend crucially on the target application, as detailed below (response to R3-C2). We thus regard this paper as a seed for further research dedicated to best practice recommendations for specific applications.

- RE “MICCAI board has taken some measures” but risk remains that “type of action remains isolated”: We have chosen the MICCAI society as a starting point because 50% of the competitions (tasks) analyzed were organized in the scope of MICCAI. As a result of our project, the MICCAI challenge working group is now in contact with both the American College of Radiology (ACR) and the Radiological Society of North America (RSNA) to develop a joint strategy for next-generation challenges in the field of medical image processing. Furthermore, future organizers of the IEEE ISBI conference (2nd most commonly used platform for organizing biomedical imaging challenges) have contacted us to transfer the structured challenge submission system that was put in place for MICCAI 2018 to ISBI.
- RE “manuscript should be published by the different societies that are stakeholders of challenges in the form of a position paper simultaneously published in their journals”: We think that a high-impact and society-neutral paper will help us with our mission of bringing biomedical image analysis challenges to the next level. We further believe that the impact of this paper can be expected to go beyond the field of biomedical image analysis and are thus convinced that publication in a general journal that reaches a broad audience is the best strategy. Furthermore, we will address this comment by contacting the editors of prominent journals in the field (e.g. IEEE TMI, Medical Image Analysis). Based on our work, we will compile a document as a guideline for authors and reviewers of biomedical challenge papers.

R3-C2: I would have also appreciated that the authors go more into strong recommendations, in particular concerning the metrics and the aggregation method(s). They rightly spot that the order of data aggregation and the metrics can influence the scoring. I would welcome a recommendation on which metrics to use and which ones to ban, for example why and when use the Hausdorff distance (HD) or the 95% variant (HD95), what are the advantages/caveats of the Jaccard index or why one should never use a given metrics. The same would be necessary for the aggregation methods.

Response:

- RE metrics: Choosing the right metric for a given domain is complex and depends crucially on the target application. Hence, we do not think that it is feasible to ban/recommend specific metrics without taking the specific medical or biological context into account. The paper by Taha and Hanbury 2015, referenced in Supplementary References [125], nicely

illustrates the complexity of the problem in the context of 3D medical image segmentation.

- RE ranking schemes: To address the reviewer's comment, we have generated a document that presents and compares three different methods for computing ranks: (1) Metric-based aggregation, the most widely used method, (2) case-based aggregation, the 2nd most commonly used ranking scheme, and (3) significance ranking, a scheme that the authors consider a promising alternative when requiring straightforward missing value handling and the natural assignment of identical ranks to algorithms that show only marginal differences.

R3-C3: In this context where the authors would indeed make recommendations rather than observations, I would suggest to drop the catchy (in the sense of sensationalistic) part of the title ("Is the winner really the best?") and spell out the purpose of the paper in the title "A critical analysis of ... and recommendations".

Response: We believe that an attention-grabbing and readily memorable title is important for an article of this nature. We changed the title to a less sensational version.